# Epigenetic Regulation: A Link between Inflammation and Carcinogenesis

**DOI:** 10.3390/cancers14051221

**Published:** 2022-02-26

**Authors:** Bianca Vezzani, Marianna Carinci, Maurizio Previati, Stefania Giacovazzi, Mario Della Sala, Roberta Gafà, Giovanni Lanza, Mariusz R. Wieckowski, Paolo Pinton, Carlotta Giorgi

**Affiliations:** 1Department of Medical Sciences, Section of Experimental Medicine, University of Ferrara, 44121 Ferrara, Italy; crnmnn@unife.it (M.C.); stefania.giacovazzi@edu.unife.it (S.G.); mario.dellasala@unife.it (M.D.S.); paolo.pinton@unife.it (P.P.); 2Laboratory of Technologies for Advanced Therapy (LTTA), Technopole of Ferrara, 44121 Ferrara, Italy; 3Department of Translational Medicine, Section of Human Anatomy and Histology, University of Ferrara, 44121 Ferrara, Italy; prm@unife.it; 4Department of Translational Medicine, University of Ferrara, 44121 Ferrara, Italy; gfr@unife.it (R.G.); lng@unife.it (G.L.); 5Anatomic Pathology Unit, University Hospital of Ferrara, 44124 Ferrara, Italy; 6Laboratory of Mitochondrial Biology and Metabolism, Nencki Institute of Experimental Biology, 02-093 Warsaw, Poland; m.wieckowski@nencki.edu.pl; 7Maria Cecilia Hospital, GVM Care & Research, 48033 Cotignola, Italy

**Keywords:** epigenetics, carcinogenesis, inflammation, tumor microenvironment

## Abstract

**Simple Summary:**

Epigenetics encompasses all the modifications that occur within cells that are independent of gene mutations. The environment is the main influencer of these alterations. It is well known that a proinflammatory environment can promote and sustain the carcinogenic process and that this environment induces epigenetic alterations. In this review, we will report how a proinflammatory microenvironment that encircles the tumor core can be responsible for the induction of epigenetic drift.

**Abstract:**

Epigenetics encompasses a group of dynamic, reversible, and heritable modifications that occur within cells that are independent of gene mutations. These alterations are highly influenced by the environment, from the environment that surrounds the human being to the internal microenvironments located within tissues and cells. The ways that pigenetic modifications promote the initiation of the tumorigenic process have been widely demonstrated. Similarly, it is well known that carcinogenesis is supported and prompted by a strong proinflammatory environment. In this review, we introduce our report of a proinflammatory microenvironment that encircles the tumor core but can be responsible for the induction of epigenetic drift. At the same time, cancer cells can alter their epigenetic profile to generate a positive loop in the promotion of the inflammatory process. Therefore, an in-depth understanding of the epigenetic networks between the tumor microenvironment and cancer cells might highlight new targetable mechanisms that could prevent tumor progression.

## 1. Introduction: The Complexity of Carcinogenesis

Carcinogenesis is a complex multistep process that leads to the onset of tumors in normal tissue in vivo. Indeed, normal cells can become transformed by accumulating several gene mutations. This carcinogenic process can be divided into three different parts: initiation, promotion, and progression. Carcinogenesis is usually initiated by the progressive accumulation of sporadic mutations that normally accumulate during the lifespan of a cell. Most of these mutations are supposed to have no relevant role in tumor illness and can also be found in normal tissues. However, a limited number of mutations, called “driver” mutations, confer a growth advantage to the cell, which in turn, through a modest but significant increase in the replication rate, can further prompt the appearance of other driver mutations, leading to cancer promotion. This process can require years, or decades, to originate a primitive tumor. Generally, mutations in the metastatic cancer are not particularly different from those of the primitive tumor, raising the problem of which genetic alteration can support the metastatic phenotype. In addition, somatic driver mutations affect a very limited number of genes and intracellular pathways [1]. These findings, together with the increasing consciousness that DNA expression and cellular phenotype are regulated by other factors in addition to gene sequence, prompted the exploration of other mechanisms that could induce cancer onset, such as epigenetic regulation. In contrast to gene mutation, most epigenetic regulation modifies gene expression without permanent changes in the genomic sequence. More importantly, epigenetic modulation is reversible and faster regulated than the establishment of gene mutation, namely genomic evolution [2]. Substantially, epigenetics controls the interactions among DNA, RNA, and the nucleosome with modifying proteins and without inducing gene mutations. One of the possible epigenetic controls is the modulation of transcription factors by limiting their accessibility to the DNA filament and determining which genes will be expressed. The other main epigenetic events are DNA and RNA methylation–demethylation and chromatin remodeling by histone acetylation–deacetylation. Because epigenetic modulation is centered on reversible interactions with different structures, it is also modifiable by environmental influences, aging, and drugs. This aspect is of particular interest because, although DNA alterations are not amenable to pharmacotherapy, there is a growing number of small molecules that can be used as epigenomic drugs for anticancer purposes and that are less toxic than traditional chemotherapy [3,4]. In fact, promising results have been recently reported on the combined use of DNA methylation inhibitors and histone deacetylase inhibitors in clinical trials [5]. The influence of the surrounding environment in determining the epigenome of a cancer cell once more highlights the importance of the tumor microenvironment (TME) in defining the initiation and progression of tumor growth. The TME consists of a supporting structure and cells that surround the tumor foci, creating a niche that can either foster or suppress tumor growth. In the TME, inflammatory cells play a pivotal role by releasing different stromal factors, such as chemokines and cytokines, which modulate carcinogenesis. Therefore, it is not surprising that the crosstalk between the TME and cancer cells is strictly mediated by epigenetic modifications [6]. In this review, we will focus on describing how epigenetic modifications regulating inflammation are able to influence cancer onset and progression.

## 2. Overview of Epigenetic Modifications

The mechanisms determining epigenetic modifications can be divided into four groups: DNA methylation, RNA methylation, histone posttranslational modifications, and the broad family of epigenetic regulators constituted by noncoding RNAs (ncRNAs) (Figure 1).

### 2.1. DNA Methylation

Epigenetic regulation can act at different levels in the flux of information from DNA to the cellular periphery. At the first level of transcriptional control, we find DNA methylation. DNA methylation occurs directly at the DNA level and allows a reduction in DNA transcription without affecting its gene sequence [7]. DNA methylation implies the covalent modification of a nucleotide and, in mammals, typically but not exclusively involves the methylation of cytosine at position 5 of the pyrimidine ring (5mC) [8]. Interestingly, 5mC is preferentially located in CpG dinucleotides, and is normally concentrated in large clusters named CpG islands, of which there are reported to be approximately 29 million in the human genome, 70–80% of which are methylated in somatic cells [9]. Non-CpG methylation occurs more frequently in human embryonic or induced pluripotent stem cells [10]. As a general rule, DNA methylation behaves as a negative regulator, reducing promoter accessibility and, consequently, the transcription of downstream genes [11]. DNA hypomethylation shows a clear, positive correlation with higher transcriptional activity. While the methylation of a promoter correlates with transcriptional repression, it has been shown that methylation on the coding gene body is related to an increase in gene expression [12].

The proteins involved in DNA methylation can be grouped into three main categories, which cover and ensure all the aspects of this regulatory process. These groups involve (i) DNA methyl transferases (DNMTs), often called the “writers”, which ensure de novo methylation (such as DNMT3A and DNMT3B) or its maintenance over time and during DNA replication (such as DNMT1); (ii) a broad set of proteins involved in 5mC recognition termed the “readers” (such as methyl-CpG binding protein 2 (MECP 2), methyl-CpG binding domain protein (MBD) 1–6, the Kaiso family, and ubiquitin-like proteins (UHRF1 and UHRF2)), which can recognize the methylation mark and can be recruited at the chromatin level mediating the silencing of the target gene; and (iii) several DNA demethylases (such as ten-eleven translocation (TET) enzymes) that are involved in the process of the removal of the methyl group, called the “erasers” [13]. As far as tumorigenesis is concerned, there are two types of changes in DNA methylation that can occur: first, the demethylation of oncogene promoters, and second, the de novo methylation of selected CpG islands [14]. Accordingly, several types of cancer present wide demethylation zones, while others, such as gliomas, cholangiocarcinomas and lymphomas, are characterized by mutations in the isocitrate dehydrogenase (*IDH*) genes, which induce hypermethylation [15]. Interestingly, de novo methylation of CpG islands is a widespread programmed process mediated by polycomb-targeting complexes, a group of epigenetic repressors that operates by recruiting the de novo methylases DNMT3A and DNMT3B in tumors [16,17,18]. Notably, the DNA methylation pattern is continuously reformed throughout human life based on the crosstalk between DNMT and TET. This indicates that DNA methylation is a dynamic activity and is therefore susceptible to environmental influences, such as nutrient availability, physical effort, illnesses, and interactions with drugs. Moreover, during aging, this fine-tuned mechanism becomes progressively unbalanced, suggesting that aging cells could represent a favorable environment, which, if hit by protumoral mutations, can more easily move toward carcinogenesis [19]. For example, abnormal hypermethylation of the MutL homologus 1 (*MLH1)* promoter, a gene involved in hereditary nonpolyposis colorectal cancer has been related to gene silencing and microsatellite instability, which in turn could initiate genetic instability in colorectal and endometrial cancers [20,21,22,23]. Interestingly, most large bowel adenocarcinomas with *MLH1* methylation display widespread methylation of the promoter of several genes, the so-called CIMP phenotype, and the BRAF-V600E mutation. The CIMP phenotype has been described in several tumor types; however, its molecular bases are poorly defined [24].

### 2.2. RNA Methylation

In addition to DNA, RNA also represents an effective target for epigenetic regulation via both methylation and ncRNA interference. RNA methylation occurs at N6-methyladenosine (m6A) and has been reported to affect the complexity of cancer progression by regulating RNA processing, nuclear export, and RNA translation. Moreover, m6A modification also occurs on ncRNAs, indicating that it controls RNA functions both directly and indirectly [25]. For DNA methylation, m6A modifications are also reversible and dynamic. Correspondingly, the major players can be categorized into (i) writers, such as different methyltransferase-like proteins (METTL3, METTL 14, METTL16), RNA binding motif protein (RMB) 15/15B, and others; (ii) readers, such as YT521-B homology (YTH) domain-containing proteins, eukaryotic initiation factor 3 (eIF3), and heterogenous nuclear ribonucleoprotein (HNRNP) protein family members; and (iii) erasers, such as alkylation repair homolog 5 (ALKBH5) and fat mass and obesity-associated protein (FTO) [6,26,27,28]. An increasing number of studies have shown that aberrant m6A modifications are closely associated with different types of cancer, such as glioblastoma, cervical and endometrial cancer, hepatocellular carcinoma, acute myeloid leukemia, breast cancer, pancreatic cancer, and prostate cancer (fully reviewed in [29]).

### 2.3. Histone Posttranslational Modifications

As previously described, DNA methylation acts directly on DNA, while histone modification implies the chemical variation of the proteins that form the nucleosome, determining whether and when specific genes will be transcribed or silenced.

Histone proteins are the targets of a broad set of chemical modifications, including acetylation, methylation, phosphorylation, ubiquitination, ADP-ribosylation, citrullination, SUMOylation, and others. All these modifications are reversible and, together with the presence of different histone isoforms, permit the existence of a broad and only partially clarified panel of different spatial chromatin rearrangements [30]. As a whole, these histone modifications constitute a truly complex chromatin-based signaling system, often referred to as histone code. This is based on changes in the charge, density, and hydrophobicity of the strands, which induce a conformational shift in the protein structure and permit the docking of specific regulatory proteins [31].

Among these modifications, the most studied is histone acetylation, which acts on the charge of the lysine residues. On the one hand, histones typically have a cumulative positive charge, which allows a strong interaction with the negatively charged DNA strand. On the other hand, acetylation occurs at the N-terminal extremity of the proteins, neutralizes the lysine charge, and reduces the histone-DNA interaction. Acetylation represents a steady state between the action of histone acetyltransferases (HATs), which transfer an acetyl group from acetyl-CoA onto lysine to form ε-N-acetyl lysine and histone deacetylases (HDACs), which remove this acetyl group. HATs are represented by many proteins, often presenting a bromodomain, which display their activity directly on histones packaged into the nucleosomes or previously in the cytoplasm on unassembled histones. Primarily, HATs target histones H3 and H4; however, H2A and B can also be acetylated on several lysine residues [32]. The classic HAT *p300* has been found to be mutated in several cancers [33], and in a similar way, PCAF, a p300/CREB binding protein (CBP)-associated factor, is negatively associated with lung or gastric cancer [34,35]. Consistently, general hypoacetylation occurs during carcinogenesis, together with altered acetylation patterns, such as for histone H4K16 or other residues [36,37,38]. Consequently, many HDAC inhibitors (HDACis), also defined as epidrugs, have been used in cancer therapy, as reviewed elsewhere [39].

Another important histone modification is methylation, which can occur on all histone proteins on the nitrogen atoms of different lysine (K) or arginine (R) residues. In addition, each residue, be it methylated, trimethylated, acetylated, or modified in a different way, admits the docking of several reader proteins involved in the packaging or unpackaging of chromatin. These processes can involve different portions of the DNA filament, influencing the accessibility of the gene sequence and of gene promoters or enhancers [40]. This activity, together with proper DNA methylation, contributes to determining cell identity and tissue enrollment [41] and their maintenance over time. Moreover, the DNA sequence and its methylation status act as factors regulating nucleosome occupancy. In fact, repetitive DNA regions enriched in methylated CpG islands have been reported to strongly affect interaction with the octamer, while unmethylated CpG islands or unmethylated transcription binding sites show the lowest occupancy levels [42]. Inactivated regions appear to be characterized by polycomb protein attachment [43,44]. In particular, enhancer of Zeste homolog 2 (EZH2) is a histone methyltransferase member of polycomb repressive complex 2 (PRC2), which normally methylates lysine 27 of histone H3 (H3K27) [45]. For this reason, its activity keeps the chromatin in a repressed state throughout the cell cycle and, together with the methylation of lysine 9 of histone 3 (H3K9) and of other laminin-associated factors, helps to transmit cell identity and tissue commitment to daughter cells. Therefore, it is not surprising that mutations in this enzyme are frequent in cancer [45]. Consistently, *EZH2* is frequently mutated in B cell lymphoma and melanoma [46]. *EZH2* somatic mutations induce hypermethylation activity on H3K27, followed by the depletion of other critical genes, leading B cells to remain in a permanent proliferative state [47].

Last, the ATP-dependent remodeling of nucleosome position is led by switch–sucrose non-fermentable (SWI–SNF) complexes, which are a large protein family of ATP-dependent chromatin remodeling complexes. This family has been regarded as having tumor suppressor activity and was found to be frequently mutated in several malignancies, including chronic and acute leukemia, lymphomas, rhabdoid tumors, and ovarian cancers [48]. In particular, AT-rich interactive domain-containing protein 1A (*ARID1A*), a member of the SWI-SNF family, was found to be mutated in gastric and pancreatic cancers and is related to breast cancer metastasis and indicative of trastuzumab resistance [49,50].

Histone lactylation is a recently studied posttranslational modification [51]. Zhang et al. found that lactic acid, already known to promote gene expression and histone acetylation [52], can directly tag lysine residues on H3, H4, H2A, and H2B histones. Histone lactylation also occurs in lung tumors and melanoma cells. Moreover, exogenous lactate decreases the HDAC content in the nucleus, HDAC activity, and chromatin methylation [52,53]. HDAC inhibition occurs at IC50 values which are not only lower than those of other pharmacological inhibitors, but also higher than reported intracellular physiological lactate concentrations. Collectively, these data suggest that lactate can potentially transduce the modifications induced by hypoxia and glucose fermentative metabolism that normally occur in the tumoral environment at the chromatin level. Further studies are needed to precisely define the exact role of lactate as an epigenetic factor in cancer onset and progression.

Altogether, the acquisition of a permissive chromatin arrangement not only predisposes cells to carcinogenesis, but also transmits them after replication to the succeeding generations, creating a cell clone with higher replication potential. Thus, inside the tumor mass, epigenetic plasticity can contribute to creating intratumor heterogeneity, which represents a valuable tool to address the variability of environmental conditions due to tumor spreading among different tissues in distant organs or during the selective conditions imposed by medical treatments.

### 2.4. Noncoding RNA: Focus on microRNA

Another level of epigenetic regulation acts directly on transcriptional activity and involves a different category of molecules, i.e., the broad class of ncRNAs. It has been estimated that only 1–2% of RNA is messenger RNA and can codify for proteins, while the remaining part consists of ncRNA. The class of ncRNA gathers several different types of RNA, such as housekeeping RNA with structural and well-characterized functions. For example, ribosomal or transfer RNA, and many different ncRNAs, present to a minor extent with a not completely clarified regulatory role. This latter group could be further divided according to size into small ncRNAs (sncRNAs, <200 nt) and long ncRNAs (lncRNAs, >200 nt) [54]. The best characterized sncRNA is miRNA, a highly conserved single-stranded RNA with ~20 nucleotides.

miRNAs play a relevant role in cancer pathogenesis through two outstanding aspects of their activity: on the one hand, the ability to control the synthesis of almost all cellular proteins, and on the other hand, the deep interdependence with the other epigenetic control mechanisms. These aspects, together with their numerosity, create a complicated network of reciprocal influences, where all the components regulate each other, and the perturbation of one or more elements often leads to some strongly dysregulated patterns. For instance, miR-15/16 are interesting examples of these interrelationships. In fact, they are listed among tumor suppressive miRNAs and are downregulated in several tumors, including chronic lymphocytic leukemia (CLL); multiple myeloma; prostate, colon, lung, and ovarian cancers; and other tumors. In CLL, miR-15/16 downregulation is due to miRNA deletion at 13q14, which occurs with high incidence in this cancer [55]. miR-15/16 deletion leads to the overexpression of several target genes depending on the specific tissue, including the anti-apoptotic factor B cell lymphoma 2 (*BCL2*) in CLL [56]; the cyclooxygenase-2 (*COX-2*) gene in colon cancer [57]; cyclin D1, the proto-oncogene protein *WNT3*, in prostate cancer [58]; *VEGFa* in multiple myeloma; and genes, such as *C-MYC* and *ALK* in other tumors [59,60]. The activity of miR-15/16 can mediate the action of P53, which in turn can both transcriptionally and posttranscriptionally control miRNAs, such as miR-34 and miR-200 [61]. Accordingly, P53 mutations, which are highly represented in a wide variety of cancers, lead to the downregulation of several miRNA families followed by an increase in the expression and activity of their target oncogenes. In particular, miR-34 controls BCL2, NOTCH, and the high mobility group AT-Hook 2 (HMGA2) in gastric cancer and MYC and MET in ovarian cancer [62,63], while miR-200 controls zinc finger E-box binding homeobox (ZEB)1, BM1, CNNB1, FN1, LEPR, and NTRK2, and inhibits cellular growth and metastasis in several cancers, including nasopharyngeal, pancreatic, and breast cancer [64,65].

Similar to mutated P53, several other oncogenes exert part of their action, dysregulating the balance of the miRNA network, by downregulating tumor suppressive miRNA. As an example, MYC is a well-known transcription factor with oncogenic activity that regulates the transcription of a broad number of miRNAs, downregulating miR-15a/16-1, miR-26a, miR-34, and let-7 family members, and consequently reducing their proapoptotic and antiproliferative effects [66]. Interestingly, the miRNA let-7 family is responsible for the ablation of MYC in Burkitt lymphoma, thus inhibiting cancer cell progression, while it targets interleukin (IL)-6 in breast cancer, the transcription factor E2F2 in prostate cancer, and the anti-apoptotic BCL, namely, BCL-XL in the liver [67]. Similarly, in Kirsten rat sarcoma (RAS) mutant pancreatic cancers, the RAS oncogene binds, through its RAS-responsive element-binding (RREB1), the promoter of the miR-143/145 family, which normally act as repressors of the same RAS and RREB1 transcription factors. Therefore, by inhibiting its inhibitors, mutated RAS strongly potentiates its own oncogenic activities [68]. The same mechanism of reciprocal inhibition occurs between the ZEB1 and miR-200 family; thus, by blocking miRNA translation, the ZEB1 and ZEB2 proteins can upregulate their expression in several cancers [69].

In addition to mutated transcription factors, several epigenetic mechanisms can alter the normal balance of miRNA in cancer. First, variation in DNA methylation strongly affects the miRNA network. For example, the loci of the miR-34, miR-124, and miR200 families are hypermethylated and epigenetically silenced in a vast number of different tumors [70,71,72]. According to the positive feedback model already seen in other dysregulated scenarios, many miRNAs include among their targets the mRNA for methylases or demethylases. This scenario can change the methylation status of numerous different gene loci, leading to the repression of tumor suppressors, the enhancement of oncogenes, and disequilibria in miRNA production, with a strong multiplicative effect on the dysregulation of cell activities.

## 3. Inflammation-Mediated Epigenetic Modifications: A Focus on Cancer Cells

The association between inflammation and cancer was first established in 1863, when Virchow postulated the tendency of cancers to develop at sites of chronic inflammation [73]. Over the years, several studies have supported the relationship between inflammation and cancer initiation and progression [74,75,76]. It is important to emphasize that, in tumorigenic processes, inflammation is a double-edged sword. Initially, inflammation is fundamental to suppress carcinogenesis, since immune cells recognize and eliminate abnormal cells; however, in the case of chronic stimuli, such as pathogen infections, inflammation can become detrimental, increasing the chance of developing carcinogenic, genetic, and epigenetic modifications within normal cells. Moreover, sustained inflammation promotes tumor progression. In fact, cancer cells activate positive loops with immune cells to maintain their inflammatory niche. In chronic inflammation-induced carcinogenesis, infections have been etiologically linked to several cancers; indeed, inflammation plays a decisive role in human papillomavirus (HPV)-induced cervical cancer and Epstein–Barr virus (EBV)-induced nasopharyngeal carcinoma [77,78,79]. Similarly, *Helicobacter pylori* (HP) infection causes chronic inflammation of the gastric mucosa, leading to gastric carcinoma [80,81]. Notably, the inflammatory process itself regulates different epigenetic mechanisms, generating a positive feedback loop that promotes carcinogenesis (Figure 2). In a gerbil model of gastric cancer, it has been reported that the expression levels of several inflammation-related genes, including C-X-C motif chemokine ligand (*CXCL*)-2, *IL-1β*, nitric oxide synthase 2 (*NOS2*), and tumor necrosis factor α (*TNF-α*), display a correlation with their methylation levels. Accordingly, the suppression of inflammation with the immunosuppressive drug cyclosporin A in HP-infected gastric mucosa cells showed a block of altered DNA methylation, indicating that the infection-associated inflammatory response induces alterations in DNA methylation [82]. In fact, infections, and more generally chronic inflammation, have been reported to play a role in approximately 25% of all cancers, including breast, colon, prostate, liver, and gastric carcinoma [80,83,84,85,86,87].

Although inflammation has been largely related to infections, it can also be promoted by exposure to various substances, such as inhalable fibers, chemicals, dust, and particulate matter. These substances promote the inflammatory process, acting as proinflammatory factors and connecting inflammation to carcinogenicity. For instance, asbestos fiber inhalation, which leads to chronic inflammation, is crucial for the development of malignant mesothelioma [88]. Accordingly, the lungs are highly influenced by lifestyle-induced inflammation, as in smokers, where profound epigenetic changes, including DNA methylation, deregulated histone acetylation, altered gene expression levels, and microRNA profiles, are the basis of a predisposition to the development lung tumors. In particular, in a nicotine-addicted mouse model, inflammation-driven changes in cytosine methylation and hydroxymethylation patterns resulted in an imbalance of DNA methylation–demethylation dynamics, which in turn gave rise to a shift in histone acetylation contributing to the initiation of lung cancer [89].

In addition to the inflammation-induced epigenetic alterations at the primary tumor site, an evident link between inflammatory pathways and epigenetic mechanisms also exists in the regulation of the metastatic process (Figure 2). Indeed, chronic inflammation not only triggers cancer development but also facilitates tumor progression. For instance, in non-small-cell lung cancer (NSCLC), IL-1β-induced epithelial–mesenchymal transition (EMT) promotes SLUG-dependent epigenetic modifications of the E-cadherin and *CDH1* promoters. Notably, upon acute IL-1β exposure, upregulated *SLUG*, a zinc-finger transcription factor, reduces activating histone modifications, such as H3K4 trimethylation (H3K4Me3) and H3K9 acetylation, and enriches repressive H3K27 trimethylation (H3K27Me3). In the continuous presence of IL-1β, SLUG accumulation leads to further enrichment of H3K27Me3 and de novo H3K9Me2/3, contributing to memorized E-cadherin suppression in EMT memory [90]. The chemical inhibition of DNA methylation, as well as the restoration of E-cadherin expression in EMT memory, also leads cells to chemotherapy-induced apoptosis. This evidence highlights the role of IL-1β-regulated chronic inflammation as a central component in carcinogenesis and metastasis, suggesting that the inhibition of this pathway may contribute to both the prevention and treatment of NSCLC. Interestingly, in mouse models of NSCLC, it has been reported that a combination of DNMT inhibitors (DNMTis) with HDACis, defined also as epidrugs, reverses tumor immune evasion and addresses T cells toward memory and effector T cell phenotypes. Thus, the inhibition of DNMT and HDAC represents a promising approach for enhancing cancer immunotherapy [91].

A further example is breast cancer, where a proinflammatory microenvironment at the distant metastatic site, particularly enriched in IL-6 and prostaglandin E2 (PGE_2_), promotes DNMT3B induction, thus altering the methylation of multiple pathways involved in cancer cell survival, apoptosis, and invasion, including signal transducer and activator of transcription 3 (STAT3), nuclear factor kappa-light-chain-enhancer of activated B cells (NFκB), PI3K/Akt, b-catenin, and Notch signaling. Hence, targeting IL-6 or PGE_2_ reduced DNMT3B induction and enhanced the efficacy of programmed cell death protein 1 (PD-1) immunotherapy in preclinical mouse models of breast cancer metastasis [92]. In addition, PGE_2_-induced DNMT3B expression and, thus, altered DNA methylation was also reported in gastric cancer, and the combined inhibition of COX-2 and DNMT inhibited gastric cancer growth both in vitro and in vivo [93]. Furthermore, DNMT3B activity has a pivotal role in IL-6-mediated octamer-binding transcription factor 4 (OCT4) expression in sorafenib-resistant hepatocellular carcinoma (HCC), highlighting *DNMT3B* as a putative therapeutic target for patients expressing cancer stemness properties or sorafenib resistance in HCC [94]. Importantly, elevated IL-1β and IL-6 levels have been found in the serum of patients with pancreatic cancer, a cancer type that strictly relies on the inflammatory process [95].

Among the inflammation-induced epigenetic modifications in cancer-related contexts, inflammation-induced miRNA alterations are being discovered. Indeed, during inflammation, the expression pattern of miRNAs is rearranged, suggesting a prominent role of extracellular miRNAs in cancer cell transcriptome reprogramming [96]. For instance, in colon cancer, high activation levels of IL-1β/NF-κB have been reported to induce the expression of miR-181a, thus promoting cell proliferation by repressing phosphatase and tensin homolog (PTEN) [97]. Similarly, in gastric cancer, the IL-1β/NF-κB axis has been demonstrated to upregulate the expression of miR-425, which can promote the growth of gastric cancer cells by negatively regulating PTEN [98]. The downregulation of microphthalmia-associated transcription factor (MITF-M) through IL-1β-induced miR-155 in melanoma cells could represent a mechanism of melanoma immune escape in an inflammatory microenvironment [99]. In addition to the upregulation of several miRNAs by IL-1β, the release of this cytokine has also been associated with the downregulation of miR-101. This event in the Xuan Wei lung cancer cell line increased the expression of *EZH2*, conferring cell proliferation and metastasizing characteristics [100]. Consistent with this result, in NSCLC cells, it has been shown that IL-1β upregulates Lin28B by downregulating miR-101, thus affecting cell proliferation and migration [101]. It has been reported that, in human colorectal cancer (CRC) cells, IL-6, another proinflammatory cytokine, activates the STAT3 transcription factor by directly repressing miR34A, leading to IL-6-induced EMT and invasion [102]. In prostate cancer cell lines, such as PC-3 and LNCaP, IL-6 administration results in the induction of miR-21, which in turn is responsible for the downregulation of the tumor suppressor programmed cell death 4 (PDCD4) [103].

All these findings support the central role of inflammation-induced epigenetic alterations in the initiation, progression, and metastasis of human cancers. Since epigenetic inheritance is reversible, its modulation may represent a valid alternative to prevent or treat cancers.

## 4. Inflammation-Mediated Epigenetic Modifications: A Focus on the Tumor Microenvironment

As mentioned above, carcinogenesis is a complex mechanism that finds its roots in the tumoral drift of specific cells supported both by genetic and/or epigenetic alterations and by a predisposed microenvironment. Specifically, epigenetic modifications also act in the TME, especially in the regulation of the immune system in the tumorigenic process. The TME consists of extracellular matrix enriched in stromal and immune cells within a continuous network of cytokines and chemokines. This scenario is susceptible to epigenetic reprogramming, resulting in the modulation of the immune response and changes in the stromal compartment (Figure 3) [6].

Starting from the cellular level, it is well known that, to escape from T cell-regulated immune surveillance, carcinogenesis needs an immune-suppressive environment. Briefly, T cells can be divided into two main groups: CD4^+^ T cells, which are highly versatile and polyfunctional, and CD8^+^ cytotoxic T lymphocytes (CTLs). Notably, CD4^+^ T cells are characterized by their ability to sense the surrounding environment and differentiate into several diverse functional subtypes in response to detected signals, becoming central coordinators of the immune response. By secreting different tumoricidal cytokines, such as interferon-γ (IFNγ) and TNFα, CD4^+^ T cells, also named CD4^+^ helper (T_helpers_), support CTLs in the disruption of primary tumor cells [104]. On the other hand, CD4^+^ regulatory T cells (T_regs_), which are characterized by their immune-suppressive activity and by the expression of the transcription factor forkhead box P3 (FOXP3), play a pivotal role in promoting tumor progression by suppressing effective antitumor immunity [105,106]. Interestingly, FOXP3 is stabilized by epigenetic modification at the *Foxp3* locus [107]. In particular, DNA demethylation of CpG motifs within this locus appears to be a crucial mechanism for the development of the stable CD4^+^ T_reg_ lineage [108]. The expression of *FOXP3* defines two major subsets of T_regs_: thymus-derived natural T_regs_ (nT_regs_) that express *FOXP3* constitutively and peripherally induced T_regs_ (iT_regs_) in which *FOXP3* expression is unstable [107]. Notably, the plasticity of iT_regs_ to go from conventional T_helpers_ to T_regs_, and vice versa, has been ascribed to epigenetic modifications based on the presence of transforming growth factor β (TGF-β) [109,110,111]. In fact, conversely to T_regs_, T_helpers_ are involved in antitumoral responses together with tumor-specific CTLs; therefore, TGF-β is, at least in part, responsible for the modulation of the inflammatory response in the TME [112,113,114,115]. Notably, TGF-β is a pleiotropic cytokine involved in both suppressive and proinflammatory immune responses. Briefly, on the one hand, TGF-β inhibits immune responses by suppressing the functions of type 1 and type 2 T_helpers_ and natural killer cells and by promoting the generation of T_regs_. However, in combination with IL-6, TGF-β promotes immune responses by inducing the generation of type 17 T_helpers_ [116,117,118,119].

Epigenetic mechanisms involving DNA methylation have also been reported to be fundamental for CTL differentiation. Specifically, it was reported that the shift from the methylation to the demethylation state of biologically relevant gene promoters is required for the transition from naïve CTLs into effector cells, thus endowed with antitumoral effects [120]. In this regard, the transition between naïve and activated immune cells is regulated by different checkpoint proteins that are expressed both on the surface of T cells and of target cells, such as cancer cells. These immune checkpoints can be regulated by changes in the DNA methylation pattern and enrichment of methylated histone marks in the promoter regions induced by the TME [121]. The most studied and characterized are cytotoxic T-lymphocyte-associated protein 4 (*CTLA-4*), *PD-1*, lymphocyte-activation gene 3 (*LAG-3*), T cell immunoglobulin and mucin domain 3 (*TIM3*), B- and T-lymphocyte attenuator (*BTLA*), and T cell immunoreceptor with Ig and ITIM domains (*TIGIT*) [122]. For instance, it has been described that, in breast and colorectal cancer, *PD-1*, *CTLA-4*, *TIM3*, and *TIGIT* are hypomethylated both at the DNA level and in the histones of their promoters compared to healthy tissue [123,124].

Likewise, macrophage activation is also regulated epigenetically. For instance, *HDAC3* was reported to be crucial in activating the inflammatory gene expression program in lipopolysaccharide (LPS)-stimulated macrophages [125]. Additionally, an enzyme belonging to the TET family of enzymes, namely TET2, specifically represses the transcription of *IL-6* by recruiting HDAC2 and resolving inflammation [126]. Another study investigated the contribution of DNMT3B to macrophage polarization and inflammation and showed that it regulates the methylation of the peroxisome proliferator-activated receptor *(PPAR)γ1* promoter, a critical regulator of alternative macrophage activation. Briefly, DNMT3B knockdown stimulates macrophage polarization to an alternatively activated M2 (anti-inflammatory) phenotype, thus correlating DNMT3B activity in macrophages with lower inflammation. Accordingly, DNMT3B overexpression revealed its important role as a negative regulator of M2 macrophage polarization, confirming the important role of the methylation state in the modulation of the inflammatory response [127]. In addition, Zhang et al. [51] showed a positive correlation between ARG1 expression and histone lactylation in tumor-associated macrophages (TAMs) isolated from melanoma and lung tumor cells and suggested, as previously proposed by Colegio et al. [128], that lactate can regulate TAM polarization toward an M2-like phenotype and, consequentially, have an important role in tumor growth.

Cytokines and chemokines are critical key points for the correct communication between different immune cells and for their recruitment to the TME, and their promoters are often epigenetically regulated.

For instance, the 5′ region of the *IL-4* locus is specifically demethylated during type 2 T_helper_ differentiation, promoting high levels of cytokine production [129]. Accordingly, different studies supported that IL-4 production is epigenetically regulated by DNMTs and HDACs [130,131]. Similarly, 5-azacytidine (5-AZA), a DNMT inhibitor, is able to induce the production of IFNγ by type 2 T_helper_ murine cells, which are unable to secrete this cytokine under normal conditions [132]. Furthermore, the absence of DNMT1 increases the expression of other cytokines, particularly IL-4, IL-5, IL-10, and IL-13, in CTLs [133]. However, these latter cytokines, together with other chemokines, are epigenetically suppressed in many cancers. As an example, trimethylation at H3K27 represses the production of CXCL9 and CXCL10 in ovarian cancer, establishing an immune-suppressive TME [134], while DNMT1 is responsible for the diminution of CXCL12 in osteosarcomas, resulting in reduced CTL recruitment at the cancer site [135]. These findings indicate how the use of specific epidrugs might reduce cancer progression by modulating inflammatory cytokine expression.

In addition to cytokines and chemokines, extracellular vesicles (EVs) have an important role in mediating communication between different immune cell types within the TME [136]. EVs are named according to their dimension and, depending on whether they are of intracellular or cellular origin, they can be defined as exosomes, macrovesicles, apoptotic vesicles, and oncosomes. EVs contain DNA, mRNA, ncRNAs, reactive oxygen species (ROS), cytokines, and chemokines, and are directly connected with carcinogenesis. In fact, their contribution starts with the generation of a protumoral environment and ends with the coordination of chemoresistance and metastatic processes [137,138,139,140,141]. For instance, exosomes secreted by ovarian cancer cells induce the production of IL-6, IL-1β, and TNF-α within monocytes through toll-like receptor (TLR) activation, which in turn activates the STAT3 pathway in immune cells, supporting the suppression of antitumorigenic inflammation [142]. Moreover, two closely related miRNAs, miR-146a, and miR-146b, known to be present in exosomes [143], are important regulators of the immune response. miR-146a/b are expressed in response to high levels of the inflammatory mediators, i.e., IL-6 and IL-8, and regulate TLRs and cytokine signaling through a negative feedback loop [144,145]. Interestingly, the administration of mesenchymal stem cell (MSC)-derived exosomes containing miR-146/b significantly reduced glioma xenograft growth in a rat model of primary brain tumors [143]. Thus, it is possible to hypothesize that miR146a/b are involved in tumor growth modulation by decreasing the secretion of inflammatory cytokines. In conclusion, epigenetic modifications have a central role in the modulation of inflammation within the TME and can represent a possible target for the treatment of cancer.

## 5. Anti-Inflammatory and Epidrug Applications in Cancer Therapy

In the previous paragraphs, we described how different inflammatory stimuli can lead to epigenetic modifications and how epigenetic alterations in the immune system result in increased inflammation. Since both of these situations translate into a favorable tumorigenic environment, the application of drugs targeting one or both of these pathways has become promising in cancer therapy.

Interestingly, different anti-inflammatory drugs have also been shown to act at the epigenetic level. As an example, nonsteroidal anti-inflammatory drug (NSAID) administration resulted in the suppression of cancer progression through the regulation of HDAC expression. Particularly, in a mouse model of colitis-associated colon cancer, aspirin promoted a reduction in H3K27 acetylation in inducible nitric oxide synthase (iNOS), TNF-α, and IL-6 promoters, leading to a dramatic suppression of both mRNAs and proteins [146]. Furthermore, in COX-1-positive ovarian cancer cells, aspirin enhanced the effects of romidepsin, an HDACi epidrug, through the augmentation of P21 expression, thus potentiating the inhibition of tumor growth [147]. Further support for the role of NSAIDs in epigenetic-mediated cancer modulation comes from studies reporting that the prolonged use of ibuprofen also correlates with a reduction in the risk of developing several cancers [148]. Notably, in vitro and in vivo studies revealed that ibuprofen diminished cancer cell metastasis, stemness properties, and cancer cell chemoresistance through a reduction in *HDAC* and histone demethylase KDM6A/B expression in a COX2-dependent manner [148]. In A549 lung cancer cells, MDA-MB-231 breast cancer cells, and HepG2 liver cancer cells, ibuprofen inhibited inflammation-related stemness genes, including *IL-1α*, *IL-1β*, *ICAM3*, *CCL16*, *TRAF6*, *PDE3A*, *PRTN3*, *NFκB1*, *IκBκB*, and *BCAR1* [148].

At the same time, epidrugs that have been designed for the modulation of epigenetic mechanisms have shown anti-inflammatory properties. For example, it has been shown that the combination of the HDACi MS-275 and resveratrol, a sirtuin 1 activator, reduced inflammation in vivo in ictus models by inhibiting microglia–macrophage activation [149]. Moreover, the combination of trichostatin A and 5-AZA, an HDACi and a DNMTi, respectively, mitigates inflammation-induced pyroptosis and apoptosis in acute lung injury by inhibiting the activity of IL1β, caspase 3, caspase 9, and caspase 11 in bone-marrow-derived macrophages [150].

In recent decades, a new class of epidrugs targeting the bromodomain (BRD) and extraterminal family (BET), named BET inhibitors (BETis), have been developed for cancer treatment [151]. Briefly, BET are BRD-containing proteins involved in gene expression regulation through histone recognition and modification, chromatin remodeling, and transcriptional machinery regulation [152]. BET performs this regulatory activity by recognizing the e-N-acetylation of lysine residues (Kac) on histone tails [151]. Their inhibitors, BETis, have been successfully used for the in vitro and in vivo treatment of different cancer types, such as nuclear protein in testis (NUT) midline carcinomas and hematological malignancies [151,153]. As an example, recent studies have identified the application of two BETis, I-BET151 and ABBV-075, which induce apoptosis in MLL-fusion leukemia, acute myeloid leukemia, non-Hodgkin lymphoma, and multiple myeloma cells [154,155]. Fascinatedly, these compounds have strong anti-inflammatory proprieties. The BETi I-BET151 suppresses the expression of TNF-α, IL-1β, and TLR ligands, resulting in a reduction in the proliferation rate and immune cell recruitment capacity in rheumatoid arthritis synovial fibroblasts [156]. High-throughput screening revealed that both I-BET151 and Ro 11-1464, another BETi, upregulate the mRNA expression of the endogenous tumor suppressor protein CEBPD and suppress *IL-6* and *CCL2* gene expression in cultured macrophages [157], reinforcing their anti-inflammatory role.

All these studies further validate the idea that tumorigenesis and inflammation are linked by epigenetics. Therefore, anti-inflammatory drugs combined with epidrugs represent a promising strategy in the prevention of cancer progression.

## 6. Conclusions

Epigenetic modifications occur in response to environmental changes and play a fundamental role in gene expression following environmental stimuli. As described, the inflammatory process plays a key role in the regulation of the initiation and progression of carcinogenesis. At the same time, carcinogenesis is responsible for the induction of a protumorigenic process. Interestingly, both these processes can be connected by a common denominator: they are controlled by epigenetics. Since epigenetic modifications are reversible and highly influenced by the surrounding environment, it is fundamental to deeply elucidate how these mechanisms promote the advancement of cancer growth and diffusion. For instance, many studies have demonstrated how aberrant epigenetic modifications play a key role in cancer incidence through the generation of specific methylation patterns [158,159]. Notably, due to their reversible regulation, epigenetic modifications are a promising target for cancer therapy. Indeed, epidrugs could directly target the alterations within the tumor core and those occurring in the TME. Moreover, with the advent of high-throughput epigenome mapping technologies, it will be of priority interest to study the epigenomic map of patient-derived cancer cells to find a more suitable therapeutic approach. Finally, it will be important to identify an “inflammatory identity card”, which will describe both the immune cells and the inflammatory cytokines present within the TME, allowing the use of patient-tailored synergic therapies.

## Figures and Tables

**Figure 1 cancers-14-01221-f001:**
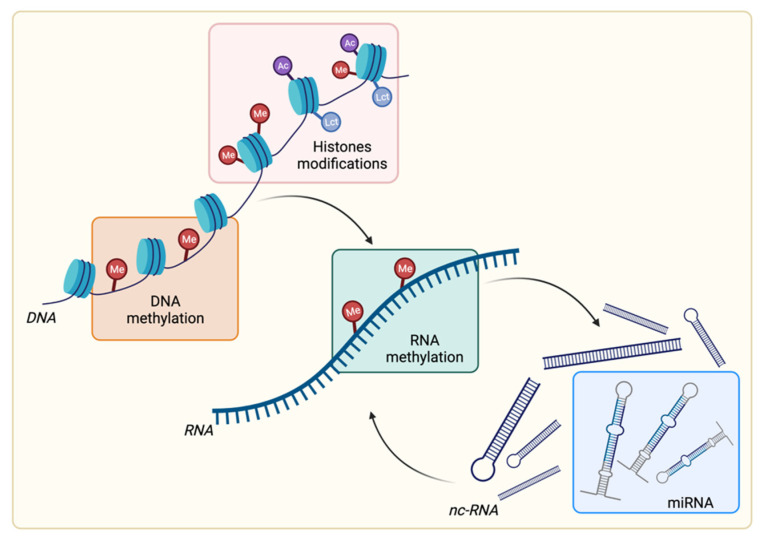
Overview of epigenetic modifications. Schematic representation of the four major epigenetic modifications, DNA methylation, histone modifications, RNA methylation, and miRNAs, as part of the broad family of noncoding RNAs (ncRNAs). Me: methylation; Ac: acetylation, Lct: lactylation. Created with BioRender.com.

**Figure 2 cancers-14-01221-f002:**
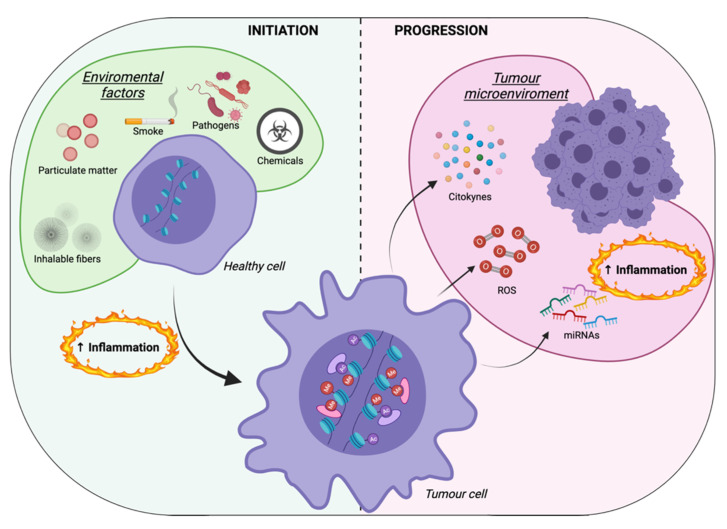
Role of inflammatory-driven epigenetic alterations in cancer initiation and progression. Chronic exposure to proinflammatory environmental factors, such as inhalable fibers, particulate matter, smoke, pathogens, or chemicals, promotes epigenetic alterations that trigger cancer development. At the same time, proinflammatory signals, such as miRNAs, ROS, and cytokines, are released by tumor cells within the tumor microenvironment and are responsible for tumor progression and metastasis. ROS: reactive oxygen species; Me: methylation; Ac: acetylation. Created with BioRender.com.

**Figure 3 cancers-14-01221-f003:**
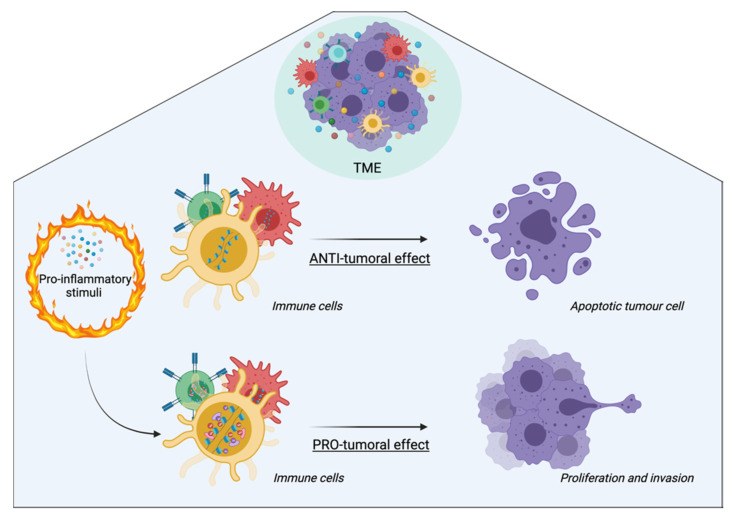
Role of inflammatory-driven epigenetic alterations in the TME. In the early stages of tumorigenesis, immune cells drive an antitumoral response. Within the TME, there is a continuous network of cytokines and chemokines that modulates both the recruitment and the activity of immune cells. In fact, proinflammatory stimuli (such as LPS) drive epigenetic modifications within immune cells, thus prompting tumor growth. Me: methylation; Ac: acetylation. Created with BioRender.com.

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
