# Peer review of "Epigenetic Regulation: A Link between Inflammation and Carcinogenesis"

_cancers, 2022, doi:10.3390/cancers14051221_

Round 1

Reviewer 1 Report

Substantial evidence has revealed that epigenetics plays a critical role in cancer development. Work by Vezzani et al. discussed the role of inflammation in carcinogenesis through epigenetic regulation. The authors provided an overview on epigenetic modifications, followed by focused discussions of inflammation-mediated epigenetics on both tumor cells and tumor microenvironment (immune cells). The manuscript is well-organized and clearly presented. My specific comments are as follows:

Tumor cells generate a large amount of lactic acid explained by Warburg effect. Interestingly, recent studies showed that lactate directly participates in epigenetic regulations of  histone lactylation by transferring lactyl group to lysine residues in histones (PMID: 31645732). In addition, another study further showed that macrophages HMGB1, a non-histone nuclear factor, could also be lactylated (PMID: 34363018). The role of lactate in regulating polarization of tumor-associated macrophages has been reported (PMID: 25043024). Is it possible that histone lactylation plays a similar regulatory role as acetylation in tumor epigenetics? This could be discussed. 

Author Response

Reviewer 1

Substantial evidence has revealed that epigenetics plays a critical role in cancer development. Work by Vezzani et al. discussed the role of inflammation in carcinogenesis through epigenetic regulation. The authors provided an overview on epigenetic modifications, followed by focused discussions of inflammation-mediated epigenetics on both tumor cells and tumor microenvironment (immune cells). The manuscript is well-organized and clearly presented.

We thank this reviewer for the time spent reading our manuscript and for his/her very positive feedback, which is much appreciated.

My specific comments are as follows:

  • Tumor cells generate a large amount of lactic acid explained by Warburg effect. Interestingly, recent studies showed that lactate directly participates in epigenetic regulations of  histone lactylation by transferring lactyl group to lysine residues in histones (PMID: 31645732). In addition, another study further showed that macrophages HMGB1, a non-histone nuclear factor, could also be lactylated (PMID: 34363018). The role of lactate in regulating polarization of tumor-associated macrophages has been reported (PMID: 25043024). Is it possible that histone lactylation plays a similar regulatory role as acetylation in tumor epigenetics? This could be discussed. 

We thank the reviewer for this advice. We have included a brief description of histone lactylation in lines 214-225 and added a part on its regulation of TAMs in lines 484-488. We have also modified Fig. 1 including this epigenetic mechanism.

Reviewer 2 Report

Epigenetic regulation: a link between inflammation and carcinogenesis by Vezzani et al.  aims to report how an inflammatory micro-environment is related to cancer with respect to the epigenetics. Although the review can be relevant to the prevention and treatment of cancer, if the link between epigenetics networks in tumour microenvironment and cancer cells are properly highlighted, however, the manuscript despite having sufficient references have failed to provide a proper overview of the literature cited herein. The current version needs a lot of improvement to be publishable. Following are some of the suggestions:

In ‘simple summary’ the authors mention epigenetic drift, in the abstract they suggest that ‘an in-depth understanding of the epigenetics networks between tumour microenvironment and cancer cells might highlight new targetable mechanisms that could prevent tumour progression’ and in the conclusion they used the tern epi-drugs and “inflammatory identity card”. Unfortunately, the rest of the manuscript lacks support for these terms and claims. How about include them between different sections at appropriate places in the manuscript.

The introduction does not adequately introduce the concept and background for the review. Especially the reversibility of epigenetic regulation with respect to cancer and how it can be related to inflammation. Is epigenetics relevant to the onset or initiation of cancer and promotion? Why did the author mentioned to focus only on the cancer progression (line 76,77).

In the overview on epigenetic modification the figure 1 schematic is too simplistic. Can they make this image more illustrative to show the distinction between these modifications apparent and clear? Further, the information in this section is too much and too diffuse. I would suggest to make a table of these modifications and include the cancer type, cell type, modification site, involved proteins or regulatory factor and implication or remarks.

Section 3 also has a lot of information. The text on page 7 and 8 if arranged according to cancer type as they have done in the beginning of page 8 might bring some clarity and a better understanding of these information to the reader.

There is no section 4.

In section 5. There is mention of Ibuprofen, aspirin and HDACi. Did the author find epigenetic drugs relevant to inflammation or vice versa? I would suggest to make a subsection on epi-drugs and mention in separate paragraph.

Conclusion towards the end reads like they are referring to precision medicine “synergic therapy patient tailored” also why was there a mention of epigenome mapping technology. One sees a lot of new information for the first time which was not there in the manuscript. Please provide a clear overview of all the sections and then write your conclusion based on that.

The SI was not accessible.

Author Response

Reviewer 2

Epigenetic regulation: a link between inflammation and carcinogenesis by Vezzani et al.  aims to report how an inflammatory micro-environment is related to cancer with respect to the epigenetics. Although the review can be relevant to the prevention and treatment of cancer, if the link between epigenetics networks in tumour microenvironment and cancer cells are properly highlighted, however, the manuscript despite having sufficient references have failed to provide a proper overview of the literature cited herein. The current version needs a lot of improvement to be publishable. Following are some of the suggestions:

We thank our reviewer for the time spent reading our manuscript his/her abundant comments, many of which have contributed to improving our manuscript. We have tried to address them all in the best way, respecting also the comments of reviewer 1 and 3, whom did not require major changes.

  • In ‘simple summary’ the authors mention epigenetic drift, in the abstract they suggest that ‘an in-depth understanding of the epigenetics networks between tumour microenvironment and cancer cells might highlight new targetable mechanisms that could prevent tumour progression’ and in the conclusion they used the tern epi-drugs and “inflammatory identity card”. Unfortunately, the rest of the manuscript lacks support for these terms and claims. How about include them between different sections at appropriate places in the manuscript.

We have highlighted the main inflammatory mechanisms that influence epigenetic modifications in the onset of inflammatory-induced cancer types in lines 316-321 (pathogen induced inflammation), and lines 332-344 (chemically induced inflammation). In lines 345-365 we discuss how epigenetic modification induced by IL-1beta exposure prompt the metastatic process and how the use of DNMTi and HDACi (two epi-drugs) can reduce this process. The use of DNMTi is discussed also in line 360-365. In lines 366-373, we described how a pro-inflammatory microenvironment promotes epigenetic modifications in breast cancer. In lines 494-506 we described how compounds acting on epigenetic mechanisms influence the release of cytokines by immune cells. To facilitate the comprehension, we have inserted the term epi-drugs in line 181, 362, 505,560,568, 587,602.

  • The introduction does not adequately introduce the concept and background for the review. Especially the reversibility of epigenetic regulation with respect to cancer and how it can be related to inflammation.

We thank the reviewer for this observation. Anyhow, we think that the introduction gives an overall summary of all the arguments that will be further discussed in the manuscript. In fact, the reversibility of epigenetic regulation with respect to cancer has been introduced in lines 52-58, while the relation with inflammation in lines 70-77. We think that since these topics are extensively described in the following paragraphs, increasing the information in the introduction might weigh down the reading.

  • Is epigenetics relevant to the onset or initiation of cancer and promotion? Why did the author mentioned to focus only on the cancer progression (line 76,77).

We improperly used the term cancer progression, aiming to indicate the whole tumorigenic process. We correct line 79 including cancer onset.

As described in lines 41-47, carcinogenesis starts with the accumulation of sporadic mutation, that lead tumor onset and promotion. The following accumulation of other mutations, lead to tumor progression. In all these processes epigenetics has a relevant role (as underlined in line 52-58).  In fact, in the review we cited many times how epigenetics is relevant in cancer onset and progression: lines 112-134; lines 201-205; lines 206-213; lines 226-231; lines 249-268; lines 269-284; lines 322-326; lines 337-340; lines 347-360; lines 366-382.

  • In the overview on epigenetic modification the figure 1 schematic is too simplistic. Can they make this image more illustrative to show the distinction between these modifications apparent and clear? Further, the information in this section is too much and too diffuse. I would suggest to make a table of these modifications and include the cancer type, cell type, modification site, involved proteins or regulatory factor and implication or remarks.

We thank the reviewer for this observation; however, the purpose of the cited figure is to summarize the described epigenetic mechanisms discussed in the review and which are the substrates of these modifications. The suggested table would be interesting, but it goes beyond the purpose of this review.

  • Section 3 also has a lot of information. The text on page 7 and 8 if arranged according to cancer type as they have done in the beginning of page 8 might bring some clarity and a better understanding of these information to the reader.

We thank the reviewer for this advice, we rearranged according to the type of cancer in the first part of the section. However, this section has not been deeply changed since the aim of the section 3 is to highlight the involvement of different inflammation-mediated epigenetics modifications in cancer promotion, progression and metastatization. Thus, a complete change of the order by which this section has been conceived could deviate the intent of the authors in emphasizing the crucial role of these modifications in all stages of cancer development.

  • There is no section 4.

We are sorry for the inconvenience. We corrected the typo.

  • In section 5. There is mention of Ibuprofen, aspirin and HDACi. Did the author find epigenetic drugs relevant to inflammation orvice versa? I would suggest to make a subsection on epi-drugs and mention in separate paragraph.

We have followed the suggestion of the reviewer and discussed the relevance of epi-drugs and anti-inflammatory drugs in the modulation of cancer progression from an epigenetic point of view, in a separate paragraph named “Anti-inflammatory and epi-drugs applications in cancer therapy” (lines 537-588).

  • Conclusion towards the end reads like they are referring to precision medicine “synergic therapy patient tailored” also why was there a mention of epigenome mapping technology. One sees a lot of new information for the first time which was not there in the manuscript. Please provide a clear overview of all the sections and then write your conclusion based on that.

We have explained the term “synergic therapy patient tailored” in the chapter dedicated to the epi-drugs. As for the epigenome mapping technology, this was a speculation on how this high throughput screening could be useful to the identification of the epigenetic alterations present in patients’ cancer cells, leading to the identification of idoneous epi-drugs for the treatment of the specific tumour.

  • The SI was not accessible.

We are sorry for the inconvenience, there are no supplementary information, the paragraph was a leftover of the template provided by the journal. We have removed it.

Reviewer 3 Report

The review is comprehensive and well written on the role of epigenetic modifications in cancer.

found some English language corrections so it would be better to get this corrected before publishing it.

The description about TGFb as pro-inflammatory in this context is not correct. It is anti-inflammatory as it promotes Treg cells which inhibit anti-Tumor immunity.

TGFb acts as pro-inflammatory in the presence of IL-6 which drives Th17 differentiation. it should be corrected with appropriate references.

There are few drugs that target bromodomain and seem to be promising. Its good to expand on those.

IFNg secretion by T helper cells Th2 seems to be not correct as Th2 cells don't secrete IFNg. Th1 cells secrete IFNg.

Author Response

Reviewer 3

The review is comprehensive and well written on the role of epigenetic modifications in cancer.

We thank this reviewer for the time spent reading our manuscript and for his/her very positive feedback, which is much appreciated.

  • found some English language corrections so it would be better to get this corrected before publishing it.

We have sent the manuscript to an editing service to be checked for the English.

  • The description about TGFb as pro-inflammatory in this context is not correct. It is anti-inflammatory as it promotes Treg cells which inhibit anti-Tumor immunity. TGFb acts as pro-inflammatory in the presence of IL-6 which drives Th17 differentiation. it should be corrected with appropriate references.

We agree with the reviewer. We have added explanatory sentences on lines 448-455, adding respective references.

  • There are few drugs that target bromodomain and seem to be promising. Its good to expand on those.

We followed the reviewer’s suggestion and added a part describing the few drugs targeting the bromodomain in the new paragraph named “Anti-inflammatory and epi-drugs applications in cancer therapy” in lines 568-585.

  • IFNg secretion by T helper cells Th2 seems to be not correct as Th2 cells don't secrete IFNg. Th1 cells secrete IFNg.

We agree with the reviewer, Th2 do not secrete IFNg, but what we reported in our review refers to a specific experiment in which the use of 5-AZA on Th2 was able to activate IFNg secretion due to modifications in the methylation pattern of the promoter. We have added a sentence to make the concept clearer. 

Round 2

Reviewer 2 Report

It can be accepted now